# The Association between Korean Clinical Nurses’ Workplace Bullying, Positive Psychological Capital, and Social Support on Burnout

**DOI:** 10.3390/ijerph182111583

**Published:** 2021-11-04

**Authors:** Seong-Ryeol Bae, Hyon-Joo Hong, Jin-Joo Chang, Sung-Hee Shin

**Affiliations:** 1Eunpyeong St. Mary’s Hospital, The Catholic University of Korea, Seoul 21431, Korea; red-cross@hanmail.net; 2National Center for Mental Health, Seoul 04933, Korea; hhong0325@gmail.com; 3College of Nursing Science, Kyung Hee University, Seoul 02447, Korea

**Keywords:** burnout, bullying, social support

## Abstract

Recurring shortages of nursing peoplepower in recent Korean society have impacted nursing organizations with burnout accounting for a major part of nursing staff turnover. Thus, we studied the associations between workplace bullying, positive psychological capital, and social support and whether they predict nursing burnout. We used hierarchical regression analysis to observe changes in influencing factors by sequentially entering general traits, workplace bullying, positive psychological capital, and social support from 166 clinical nurses at two hospitals. The analysis showed that being female (β = 0.18), working three shifts (β = 0.40), workplace bullying (β = 0.24), and positive psychological capital (β = −0.28) were predictors of burnout (*F* = 11.25, *p* < 0.001), explaining 44.5% of the variance. An analysis of the correlations between burnout, workplace bullying, positive psychological capital, and social support revealed that workplace bullying was positively correlated with burnout (*r* = 0.36, *p* < 0.001), and positive psychological capital (*r* = −0.49, *p* < 0.001) and social support (*r* = −0.37, *p* < 0.001) were negatively correlated with burnout. Thus, the higher positive psychological capital within an organization, the lower the level of burnout, suggesting that organizations should consider education programs to promote positive psychological capital. In addition, healthy organizational culture should be promoted by monitoring workplace bullying.

## 1. Introduction

Recurring shortages of nursing peoplepower in recent Korean society have impacted nursing organizations with burnout accounting for a major part of nursing staff turnover [1]. Burnout is defined as a physical, mental, and emotional syndrome that includes a negative self-concept and working attitude and loss of interest in patients and is experienced by staff members under stress due to frequent human relationships for an extended period [2].

Although the factors related to the burnout of nurses vary, violence in the workplace has been highlighted in recent studies, where a significant correlation of burnout with bullying in the workplace was shown [3,4]. Workplace bullying entails repeated emotional and physical suffering that persists for more than 6 months, either daily or weekly, from colleagues or supervisors at work. Furthermore, it can be seen as a serious form of interpersonal conflict [5]. Specifically, it refers to negative actions such as constant criticism and humiliation in the workplace, suggestions encouraging resignation, excessive monitoring of work, excessive workload, and sudden anger [6].

In recent years, workplace bullying as a factor of burnout has been the subject of multiple studies abroad. Within Korea, the topic has been pursued in the field of education since 2000 [7] and research has recently started to focus on nurses.

In a study of new nurses in Massachusetts, U.S., 31.0% of respondents had experienced bullying [8] and 27.3% of Washington State nurses reported having experienced bullying within the past 6 months [9]. A survey of Korean clinical nurses also showed that 19.0% of participants had experienced at least two different kinds of daily or weekly bullying from colleagues or bosses during the past 6 months [10]. This is three to five times higher than in the service industry (4.1%) or the electronics and steel industry (5.8%) [11]. Additionally, 23.0% of South Korean nurses are victims of malicious torment at work, and it is reported that bullying is so prevalent in the workplace that more than 65.0% of South Korean nurses experience bullying at least once in their career [7].

Although many nurses in Korea experience bullying in the workplace, they choose to ignore or avoid the problem as it is difficult to handle on an individual level [6]. Consequently, legislation on bullying in the workplace has been proposed five times in Congress in the form of amendments to the Labor Standards Act since 2013, but it has yet to be passed [11].

Workplace bullying results in physical and mental health problems, as well as decreased job satisfaction, lower productivity, poor performance, burnout, and higher turnover rates [12,13]. As workplace bullying may result in negative consequences for nurses’, medical facilities’, and patients’ safety [14], it is critical that effective measures be taken to prevent or reduce the exhaustion of clinical nurses. Such measures are also important for the management of nursing resources and nursing quality [15] as well as for hospital management and administration.

On an individual level, positive organizational behavior can be measured, developed, and efficiently managed to improve organizational performance and reduce burnout [16]. Originating from psychology, the concept of affirmative, organizational behavior is closely tied to positive psychological capital, which comprises sub-concepts such as self-efficacy, hope, optimism, and resilience. While each sub-concept has its own degree of influence, they are far more explanatory and effective when measured and interpreted together [17]. This emphasizes the importance of the positive psychological values of an individual, which may produce more than capital in the traditional sense could contribute [16,18]. In addition, it provides empirical evidence that positive organizational behavior contributes to the stability and development of the organization by improving organizational culture and staff performance [19,20,21].

Similarly, a study of organizational culture and workplace bullying among Korean clinical nurses showed that bullying in the workplace decreased in organizations that emphasized a relationship-related culture and endorsed flexibility [22]. This shows that social support mitigates the stress an individual experience from interpersonal relationships and positively affects individual outcome variables [23], suggesting that social support from supervisors and colleagues can reduce burnout [24].

Research on burnout in Korean clinical nurses has focused on the influence of various individual relevant factors such as emotional labor [25], resilience [26], the nursing environment, sympathy fatigue and sympathy satisfaction [27], organizational commitment, and positive psychological capital [28]. However, there is a lack of comprehensive research on bullying within the workplace, regarding positive psychological capacities (an individual factor) and external organizational factors. Thus, the purpose of this study was to examine the levels of workplace bullying, positive psychological capital, and social support and analyze their relationships with burnout. To this end, this study aimed to provide data needed to manage nursing resources and care for nurses’ mental health so as to contribute to the improvement and development of nursing work.

### 1.1. Study Purpose

The purpose of this study was to determine how much the burnout of clinical nurses affects workplace bullying, positive psychological capital, and social support. Based on the above review, we identified four key areas in this study: (1) identifying the degree of burnout in relation to the demographical characteristics of Korean clinical nurses, (2) examining the level of burnout, workplace bullying, positive psychological capital, and social support among Korean clinical nurses, (3) analyzing the correlations between burnout and bullying, positive psychological capital, and social support, and (4) examining the level to which burnout is predicted by bullying, positive psychological capital, and social support among Korean clinical nurses.

### 1.2. Conceptual Framework

Previous research suggests that workplace bullying has a positive correlation with burnout [1] and a negative correlation with positive psychological capital [28] and social support [24]. Therefore, in this study, the conceptual framework to analyze the association with burnout was established by viewing workplace bullying as a negative factor leading to stress, and positive psychological capital as a personal factor (internal factor) and social support as an organizational factor (external factor) that provide positive help to individuals.

## 2. Materials and Methods

### 2.1. Participants

This study was carried out in accordance with the Code of Ethics of the World Medical Association (Declaration of Helsinki) and approved by the Institutional Review Board of C University (PIRB-00154_3-009, Date of approval: 25 March 2016). The data collection period was from 13–15 June 2016. Permission to participate in the survey was obtained telephonically and online from the nursing departments of two university hospitals in Seoul. Once permission was obtained, each hospital was visited to explain the research objectives to nurses with more than 6 months of clinical experience and who agreed to complete the questionnaire. The questionnaire was distributed directly to the survey participants and collected after being completed. The sample size was calculated using G*power 3.1.2 [29] and was determined to be 147 to maintain a significance level of 0.05, an intermediate effect size of 0.15, and statistical power of 0.90. In total, 180 questionnaires were distributed, of which 175 copies were collected, with a recovery rate of 97.2%. Excluding nine questionnaires with insufficient responses, 166 copies were used for the final analysis. Clinical nurses were asked to provide informed consent prior to participating in the study.

### 2.2. Measures

#### 2.2.1. Burnout

The Korean Version Tedium Scale was used to assess burnout [30], which was adapted from the Tedium Scale developed by Pines et al. [31]. The scale comprises 20 items and has three subscales; six items measure physical burnout, seven measure emotional burnout, and seven measure mental burnout. Example items regarding the subscale that measured burnout are: “I get tired after work (physical burnout)”, “It is annoying to perform nursing work (emotional burnout)”, and “I want to leave the nursing job (mental burnout)”. Each item was scored on a 5-point Likert scale, ranging from 1 = never felt it to 5 = always felt it. Scores ranged from 20 to 100, where higher scores indicated increased burnout. The internal reliability of the tool was shown by Cronbach’s α = 0.85 in the study by Moon [30] and 0.92 in this study.

#### 2.2.2. Workplace Bullying

To measure workplace bullying, the Workplace Bullying in Nursing-Type Inventory (WPBN-TI) [32] was used. The scale comprises 16 items on three subscales: 10 items for verbal and nonverbal bullying, four items for work-related bullying, and two items for external threats. Verbal and nonverbal bullying include being subjected to malicious rumors and unpleasant gazes, while work-related bullying involves being asked to do additional work within a tight schedule or being forced to do another’s work. Finally, external threats include requiring excessive attendance in meetings and training or being excluded from these. Example questions for the bullying in the workplace subscale included (Verbal and nonverbal bullying): “I was publicly and often ignored by the other person.”, “The other person suddenly threw something at me or around me.”. Each item was answered using a 4-point Likert scale, ranging from 1 = not at all to 4 = very much. Scores ranged from 16 to 64, where higher scores indicated higher exposure to bullying in the workplace. The original tool’s internal reliability [33] was indicated by Cronbach’s α = 0.91 and was 0.88 in this study.

#### 2.2.3. Positive Psychological Capital

The Korean Version Positive Psychological Capital tool was used [34], which was adapted from the Positive Psychological Capital tool developed by Luthans et al. [17]. The scale has 16 items on four subscales: six items for self-efficacy, four for hope, three for resilience, and three for optimism. Each item was scored using a 5-point Likert scale, from 1 = not at all to 5 = very much. Scores ranged from 16 to 80, where higher scores indicated higher psychological capital. Example questions subscale included (self-efficacy, hope, resilience, optimism): “I tend to work hard to find a solution with confidence.”, “I think there are many solutions to any problem”, “Based on my previous experience, I can cope well with difficulties at work.”, and “I always try to see the positive side of my work.”. The internal reliability of the tool was shown by Cronbach’s α = 0.91 in the study by Woo [34] and 0.93 in this study.

#### 2.2.4. Social Support

The Korean Version Social Support tool was used [35], which was adapted from the Social Support tool developed by House [36]. The scale comprises eight items on two subscales, four items for superiors’ support and four for colleagues’ support, each scored on a 5-point Likert scale, ranging from 1 = not at all to 5 = very much. Scores ranged from 8 to 40 points, where higher scores indicated a higher degree of social support. There were 36 example items that measured social support, as follows (Support of boss, Support of colleague): “If something difficult occurs while working, I can rely on my direct supervisor.” and “I have a colleague who is very close to me.”. The internal reliability of the tool was indicated by Cronbach’s α = 0.85 for superiors and 0.78 for colleagues in the study by Son and Ko [35], while in this study, Cronbach’s α = 0.84 and 0.72 for superiors and colleagues, respectively.

### 2.3. Data Analyses

Data were analyzed using the software package IBM SPSS Statistics 18.0. First, demographic characteristics of the participants, which were age, gender, clinical career, religion, marital status, position, working stations, working, harassment experience, experience of witnessing bullying, harassment severity perception, and education level, were analyzed through *t*-tests and ANOVA.

Second, the relationship between burnout, workplace bullying, positive psychological capital, and social support was analyzed with Pearson’s correlation coefficients. Third, we used hierarchical regression analysis to observe changes in influencing factors by sequentially entering general traits, workplace bullying, positive psychological capital, and social support. Fourth, the internal reliability of the tools was calculated by Cronbach’s α.

## 3. Results

### 3.1. Demographic Characteristics according to the Burnout of Participants

Participants’ demographic characteristics are shown in Table 1. Most participants were female and under the age of 25. Of the participants, most were religious and unmarried. The positions were 86.7% for general nurses, and three shifts accounted for 72.3% of the working style. Furthermore, 37.3% of participants were victims of bullying and 49.4% witnessed bullying. The severity of bullying was 22.9% ‘serious’ or ‘very serious’. There was a statistically significant difference in burnout by participants’ general characteristics in terms of age, clinical career, education level, marital status, position, working style, bullying experience, and bullying severity perception.

The level of burnout, workplace bullying, positive psychological capital, and social support among participants are shown in Table 2.

### 3.2. Correlations between Burnout and Workplace Bullying, Positive Psychological Capital, and Social Support

These correlations are shown in Table 3. Workplace bullying was positively correlated with burnout (*r* = 0.36, *p* < 0.001), while positive psychological capital (*r* = −0.49, *p* < 0.001) and social support (*r* = −0.37, *p* < 0.001) showed negative correlations.

### 3.3. Influencing Factors on Burnout of Korean Clinical Nurses

A hierarchal regression analysis was conducted to determine the explanatory power of the independent variables on burnout. A multicollinearity check before the analysis showed that the Variation Inflation Factor of all independent variables was less than 10 and that the tolerance was greater than 0.1, indicating no problems in multicollinearity. Comparatively, the results of the Durbin–Watson statistic test showed no autocorrelation between the error margins of the model, while satisfying the assumption of normally distributed residuals the final model was validated (*F* = 11.25, *p* < 0.001).

The input order of independent factors for Table 4 was gender, age, education, marital status, position, and working style, which were general characteristics that showed significant differences in burnout. The independent variables were ordered as workplace bullying, positive psychological capital, and social support gender; working three shifts, workplace bullying, and positive psychological capital accounted for 44.5% of clinic nurses’ burnout.

## 4. Discussion

The main purpose of this study was to empirically investigate the associations between workplace bullying, positive psychological capital, and social support and whether they predicted burnout while controlling for general characteristics. To do so, hierarchal regression analysis was performed, revealing that workplace bullying and positive psychological capital predicted burnout, while social support did not. That is, with more workplace bullying, burnout was higher while positive psychological capital tended to have less burnout. In this study, 37.3% of nurses reported experiencing bullying at work. In previous domestic studies of nurses, 19% of nurses reported that they had experienced bullying, in a study by Nam et al. [10], while 17.2% did so in a study by Park, Kim, and Kim [6]. In other studies, 23.0% [33] and 19.1% [22] of participants reported experiencing workplace bullying, while in foreign studies of nurses, 31% [8] and 27.3% [9] of the participants experienced bullying. Therefore, this study’s findings showed similar trends as foreign studies; however, the figures are higher than in other domestic studies. There is a discrepancy and that may well be because of perceptions of the participants, reflecting the current trends in society, hospital size, and regional and organizational culture. However, as this study was limited to two university hospitals in Seoul, it is difficult to generalize our results. There is, thus, a need for further research including a wider range of participants.

Meanwhile, in this study, 49.4% of the respondents replied having witnessed bullying. This rate was higher than that of being directly subjected to bullying. In a study by Kang, Kim, and Han [37], 66.8% of participants reported that they had witnessed colleagues being bullied, while 23.8% reported directly experiencing severe workplace abuse. Furthermore, regarding their perception of workplace bullying, 22.9% of the participants reported that it was “severe” or “very severe.” This supports Kang, Kim, and Han’s [37] research, in which 27.3% participants responded that workplace bullying was “severe” or “very severe.” In this study, the number of nurses who reported workplace bullying as “severe” or “very severe” was less than those who experienced or witnessed bullying. These results can be explained as participants displaying a passive coping reaction whereby they consider bullying as natural and stay silent or even regard bullying as part of the job [6]. Thus, periodic and systematic monitoring and intervention are required to ensure that nurses who experience bullying at work are not ignored

The mean burnout of clinical nurses was 3.33 points in this study. In a study of nurses measured by the same tools, Kim [38] reported 3.56 points, while Pines and Kanner [39] reported 3.0 points for American nurses and 3.2 points for Israeli nurses. As the degree of burnout found in this study was within the range of 3 to 4 points on the Tedium Scale proposed by Pines, Aronson, and Kafry [31], according to previous studies, it is a high figure that requires intervention. Age did not predict burnout level, but nurses who were younger and with lower careers experienced more burnout than the older group and higher careers. This is the same result as in a study Han, Yang, and Yom [3], and the work difficulties and work responsibilities experienced by nurses with low years of experience were higher than the burnout of experienced nurses. It is thought that, as clinical experience increases, work proficiency and emotional stability increase, resulting in lower burnout.

The mean score of bullying in the workplace was 2.03, with verbal and nonverbal bullying and work-related bullying each scoring 2.11 and 2.15, respectively, while external threats averaged 1.35. Furthermore, the results of this study were the same as the results of Kang, Kim, and Han’s previous studies [37] and the tool developers [7], which found that bullying was in the order of work-related bullying, verbal/nonverbal bullying, and external threats. In particular, in terms of bullying types, external threats (including physical harm) scored relatively low on the 4-point scale because bullying often appears in the form of personal insults rather than physical violence and rarely reaches the use of force [9].

The mean positive psychological capital of the participants was 3.16 points. Similarly, in studies that used the same tools, Woo [34] showed similar results with a mean score of 3.30. Comparatively, positive psychological capital in Ko, Park, and Lee’s [40] and Yun’s [41] research was somewhat different, each averaging a score of 3.94 and 3.80, respectively. The difference appears to arise from the different characteristics of the test participants. To illustrate, in this study, the majority of participants were under the age of 30 and were unmarried. In contrast, in a study by Ko, Park, and Lee [40], the rate was higher among participants older than 30, and the number of married participants was similar to the number of unmarried participants. Similarly, the differences in Yun’s [41] research can be accounted for in that the participants were nursing education participants. However, in general, the positive psychological capital of nurses was above the mean of the 5-point scale.

The mean social support of the participants was 3.47 points. On average, social support from superiors was 3.35, while social support of colleagues was 3.59. In a similar study of teachers by Son and Ko [35] that used the same tools, support from superiors was scored as 3.36 and from colleagues was 3.46. Furthermore, in a study of nurses by Han [42] that used the same tools, social support from superiors averaged 3.69, while colleague support was scored at 3.52. The studies showed that nurses’ perception of social support was higher than the mean of the 5-point scale. In particular, social support from colleagues tended to be greater than support from superiors. Analysis of the correlations between burnout, bullying, positive psychological capital, and social support revealed a positive correlation between burnout and bullying, while positive psychological capital and social support showed a negative correlation with burnout. In particular, positive psychological capital had the highest correlation among the three variables. That is, burnout is higher when there is more bullying in the workplace and is lower with increased positive psychological capital and social support. This is consistent with the studies by Yeun [1] and Han, Yang, and Yom [3], who confirmed a positive correlation between nurse burnout and bullying.

Consistent with this study, the relationship between burnout and positive psychological capital was negatively correlated in the study of Ko, Park, and Lee [40]. Similarly, Yun [41] verified that nurses with high positive psychological capital suffered fewer negative symptoms when exposed to bullying. That is, developing positive psychological capital through education and training can reduce the burnout of nurses.

The relationship between burnout and social support was found to be negatively correlated in several previous studies [24,43,44]. This suggests that although social support is an external factor in the relationship between people, it is more important to establish a relationship that helps people promote their internal positive competence by embracing their surrounding environment in a positive manner and inducing internal motivation, as well as promoting self-efficacy [23,45]. As social support may vary according to the surrounding environment, it is necessary to find a way to maximize the moderation effect of social support by developing the positive capacity of the individual.

Even though the research on the relationship between workplace bullying and burnout is still insufficient, there has been progress. Han, Yang, and Yom’s [3] research on nurses verified that workplace bullying has a direct effect on burnout, which was supported by this study’s results. Yeun [1] also showed that workplace bullying and burnout have an influence on turnover. Additionally, while Yun’s [41] claims that workplace bullying is not directly connected to turnover intentions, she does acknowledge that nurses who were subjected to bullying experienced negative physical and mental symptoms and that these symptom experiences indirectly influenced turnover intentions. She also reported that nurses with high positive psychological capital reported low negative symptom experience when subjected to workplace bullying. That is, continuous physical and mental symptom experiences caused by workplace bullying acted as a stress factor and induced burnout, leading to turnover. Comparatively, positive psychological capital deterred negative symptom experience and lower burnout. In a study of nurses by Kim et al. [46], individuals with more positive personality traits showed higher job satisfaction and lower burnout levels. Although the results of Kim et al. [46] are not directly comparable with the findings of this study, they are similar in showing how a positive sentiment can influence burnout at work. It was also found that a person with highly positive psychological capital has a higher probability of overcoming the ill-natured aspects most organizations share [17].

Whether positive psychological capital directly affects burnout requires further research. However, Lee et al.’s [28] study of nurses asserted that organizational commitment and positive psychological capital significantly impact burnout, reaching similar conclusions as this study. Han’s [42] study of clinical nurses, which applied a burnout structure model, showed similar results, with positive sentiment being the strongest explanatory factor of burnout. This study and preceding research have confirmed that positive psychological capital is an important variable that deters symptoms experienced by bullying and lowers burnout. Thus, it can be argued that it is important to develop a fundamental education and training strategy that heightens positive psychological capital and nurtures internal strength. While social support showed a significant correlation with burnout, it did not significantly predict burnout when workplace bullying and positive psychological capital were simultaneously inputted.

The results of this study agree with research that emphasizes the need to create a surrounding environment where individuals can positively perceive social support in relationships with people [23,44]. That is, to maximize the social support that acts as an external factor to the individual, a process that can enhance internal positive competence is needed. However, some studies have taken opposing views, claiming that social support can directly influence burnout by helping individuals overcome maladjustment in stressful situations [23,43]. Thus, more studies of social support are needed. In addition, organizations must develop measures to systematically monitor bullying and improve working conditions.

This study has shown support for workplace bullying and positive psychological capital predicting burnout. When the main variables of this study were entered into the model, it showed that workplace bullying led to higher burnout, while internal positive psychological capital alleviated it. Comparatively, while social support was a positive external factor that provided help to individuals, it did not significantly predict burnout. This suggests that internal factors, such as positive psychological capital, are more important than external factors. Thus, as individuals with higher positive psychological capital showed higher levels of environmental factor awareness, each institution should seek to develop educational programs to promote positive competencies. Carefully planned and continued monitoring against workplace bullying will also help create a healthier organizational culture.

The limitations of this study are as follows. First, this study did not consider the size, region, and environment of the hospital. Second, the study participants were limited to nurses from two university hospitals in Seoul. Third, when looking at the gender ratio of the sample, men had a very small ratio. According to data from the National Statistical Office in Korea in 2019, the proportion of female workers in the gender ratio of nurses was 94.9%, and the proportion of men was 5.1%, which was very small [47]. Since it reflects this Korean culture, repeated research is needed in the future. Lastly, the study’s method requires discretion when attempting to generalize the results of this study. The study introduced the main variables in the order of workplace bullying, positive psychological capital, and social support; sorted the variables into internal and external factors; and then analyzed the variation of the variables’ explanatory power on burnout. This also calls for further research.

## 5. Conclusions

The purpose of this study was to investigate the relationship among burnout, positive psychological capital, and social support, which were identified as important factors of burnout in nursing organizations. By investigating the relationships among these factors, the study aimed to provide basic data that may be used to efficiently manage human resources in nursing and bring mental stability to individual nurses, while harmonizing individual nurses’ well-being by improving nursing organizations and hospitals. The results of the study showed that workplace bullying and positive psychological capital were variables that predicted burnout. That is, more workplace bullying led to higher levels of burnout, while more positive psychological capital led to lower levels of burnout. Thus, as positive influences (positive psychological capital) lead to lower levels of burnout, nursing institutions should strive to develop training programs that heighten each individual’s positive competencies. Planned and sustained monitoring against workplace bullying is also needed to establish a healthier organizational culture. To this end, specific laws and systems are needed for effective monitoring. Based on this study’s results, the following suggestions are made. First, this study limited its participants to nurses from two university hospitals in Seoul. Thus, this research should be expanded, including a wider range of participants, who account for the size, region, and environment of hospitals, to improve its generalizability. Second, programs such as strength coaching programs and emotional management programs promoting the positive competencies of clinical nurses must be developed and their efficacy tested. Lastly, we propose future repeat studies to examine the mediating effect of positive psychological capital on the relationship between bullying and burnout in clinical nurses.

## Figures and Tables

**Table 1 ijerph-18-11583-t001:** Demographic characteristics according to the burnout of participants (*n* = 166).

Characteristics	Category	*n* (%)	M ± SD	t/F	*p*	Scheffe
Sex	Female	155 (93.4)	3.36 ± 0.59	2.02	0.045	
	Male	11 (6.6)	2.98 ± 0.72			
Age			29.8 ± 6.8	7.31	<0.001	
	≤25	60 (36.1)	3.47 ± 0.58			b
	26~30	47 (28.3)	3.40 ± 0.55			b
	31~35	29 (17.5)	3.40 ± 0.57			b
	≥36	30 (18.1)	2.89 ± 0.60			a
Clinical Career (years) ^a^			7.02 ± 6.82	7.02	<0.001	
	≤1	15 (9.0)	3.31 ± 0.59			ab
	1< ≤5	70 (42.2)	3.48 ± 0.62			b
	5< ≤10	43 (25.9)	3.41 ± 0.49			b
	>10	38 (22.9)	2.97 ± 0.58			a
Education level	Junior College Grad	59 (35.5)	3.49 ± 0.58	12.18	<0.001	b
	University Grad	79 (47.6)	3.38 ± 0.54			b
	Graduate School or More	28 (16.9)	2.86 ± 0.62			a
Religion	Religious	125 (75.3)	3.30 ± 0.63	−1.19	0.236	
	Atheistic	41 (24.7)	3.43 ± 0.53			
Marital status	Unmarried	121 (72.9)	3.44 ± 0.57	−3.91	<0.001	
	Married	45 (27.1)	3.04 ± 0.62			
Position	Junior Nurse	144 (86.7)	3.43 ± 0.56	5.87	<0.001	
	Staff or Higher	22 (13.3)	2.69 ± 0.52			
Working style	2 Shift	31 (18.7)	3.11 ± 0.53	18.48	<0.001	b
	3 Shift	120 (72.3)	3.47 ± 0.54			c
	Regular	15 (9.0)	2.64 ± 0.70			a
Bullying experience	Yes	62 (37.3)	3.48 ± 0.57	2.45	0.015	
	No	104 (62.7)	3.24 ± 0.61			
Experience of witnessing bullying	Yes	82 (49.4)	3.25 ± 0.60	1.70	0.091	
	No	84 (50.6)	3.36 ± 0.51			
Bullying severity perception	Not severe at all	31 (18.7)	3.36 ± 0.51	2.87	0.038	
	Not Severe	97 (58.4)	3.25 ± 0.62			
	Severe	34 (20.5)	3.44 ± 0.58			
	Very Severe	4 (2.4)	3.33 ± 0.61			

Note: ^a^ M ± SD.

**Table 2 ijerph-18-11583-t002:** The level of burnout, workplace bullying, positive psychological capital, and social support (*n* = 166).

Variables		M ± SD	Min	Max
Burnout		3.33 ± 0.61	1.45	4.90
	Physical burnout	3.90 ± 0.69	1.67	5.00
	Emotional burnout	2.97 ± 0.74	1.29	5.00
	Mental burnout	3.20 ± 0.64	1.00	5.00
Workplace bullying		2.03 ± 0.52	1.00	3.44
	Verbal, Nonverbal	2.11 ± 0.59	1.00	4.00
	Work related	2.15 ± 0.59	1.00	3.75
	External threat	1.35 ± 0.52	1.00	3.00
Positive psychological capital		3.16 ± 0.51	2.00	4.88
	Self-efficacy	3.10 ± 0.62	1.33	5.00
	Hope	3.26 ± 0.56	1.75	5.00
	Resilience	3.08 ± 0.58	2.00	4.67
	Optimisim	3.22 ± 0.60	1.67	5.00
Social support		3.47 ± 0.56	1.63	5.00
	From superiors	3.35 ± 0.72	1.25	5.00
	From colleague	3.59 ± 0.65	2.00	5.00

**Table 3 ijerph-18-11583-t003:** Correlation between burnout and workplace bullying, positive psychological capital, and social support (*n* = 166).

	Burnout	Workplace Bullying	Positive Psychological Capital	Social Support
r (*p*)				
Burnout	1			
Workplace bullying	0.36 (<0.001)	1		
Positive psychological capital	−0.49 (<0.001)	−0.29 (<0.001)	1	
Social support	−0.37 (<0.001)	−0.40 (<0.001)	0.44 (<0.001)	1

**Table 4 ijerph-18-11583-t004:** Influencing factors on burnout of Korean clinical nurses (*n* = 166).

Variables	Model 1	Model 2	Model 3	Model 4
	β (*p*)	β (*p*)	β (*p*)	β (*p*)
General characteristics				
Gender *	0.18 (0.010)	0.21 (0.002)	0.18 (0.004)	0.18 (0.003)
Age	−0.02 (0.905)	0.06 (0.604)	0.13 (0.251)	0.10 (0.387)
Junior college graduate ^†^	0.07 (0.651)	0.11 (0.436)	0.09 (0.466)	0.08 (0.518)
University undergraduate ^†^	−0.03 (0.840)	0.02 (0.879)	0.02 (0.909)	0.00 (1.000)
Marital status ^‡^	0.11 (0.265)	0.09 (0.358)	0.07 (0.475)	0.07 (0.421)
Position ^§^	0.20 (0.080)	0.19 (0.079)	0.12 (0.259)	0.11 (0.306)
2 Shifts ^¶^	0.16 (0.168)	0.14 (0.204)	0.15 (0.142)	0.16 (0.130)
3 Shifts ^¶^	0.37 (0.005)	0.38 (0.002)	0.41 (0.001)	0.40 (0.001)
Workplace bullying		0.33 (<0.001)	0.26 (<0.001)	0.24 (0.001)
Positive psychological capital			−0.31 (<0.001)	−0.28 (<0.001)
Social support				−0.08 (0.286)
F (*p*)	7.47 (<0.001)	10.39 (<0.001)	12.25 (<0.001)	11.25 (<0.001)
R^2^	0.276	0.375	0.441	0.445
Adj. R^2^	0.239	0.339	0.405	0.406
Adj. R^2^ change	0.239	0.100	0.066	0.001

Note: S.E (Standard Error), β (Standardized Coefficient), R^2^ (R Squared), Adj.R^2^ (Adjusted R Squared); Dummy variables: * Male, ^†^ Graduate School or More, ^‡^ Married, ^§^ Staff Nurse or Higher, ^¶^ Regular.

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
