# Peer review of "The Association between Korean Clinical Nurses’ Workplace Bullying, Positive Psychological Capital, and Social Support on Burnout"

_ijerph, 2021, doi:10.3390/ijerph182111583_

Round 1

Reviewer 1 Report

There are still a number of edits I suggest before publication. My suggestions are red in the attached file.

Author Response

Point 1: There are still a number of edits I suggest before publication. My suggestions are red in the attached file.

Response 1: Thank you so much for your feedback. I’ve marked the corrections in red. Please see below for more details. Again, thank you for your feedback.

- DC or Washington state?

(2page 6line)

In a study of new nurses in Massachusetts, U.S., 31% of respondents had experienced bullying [8] and 27.3% of Washington nurses reported having experienced bullying within the past 6 months [9].

→ In a study of new nurses in Massachusetts, U.S., 31% of respondents had experienced bullying [8] and 27.3% of Washington State nurses reported having experienced bullying within the past 6 months [9].

-..is difficult to handle...

(2page 15line)

Although many nurses in Korea experience bullying in the workplace, they choose to ignore or avoid the problem as it cannot be handled on an individual level [6].

→ Although many nurses in Korea experience bullying in the workplace, they choose to ignore or avoid the problem as it is difficult to handle on an individual level [6].

-turnover rates

(2page 20line)

Workplace bullying results in physical and mental health problems, as well has decreased job satisfaction, lower productivity, poor performance, burnout, and higher intention for turnover [12,13].

→ Workplace bullying results in physical and mental health problems, as well as decreased job satisfaction, lower productivity, poor performance, burnout, and higher turnover rates [12,13].

-also

(2page 24line)

Such measures are important for the management of nursing resources and nursing quality [15] as well as for hospital management and administration. Thus, it is necessary to seek ways to replace workplace bullying with positive factors and improve the organization’s performance by 25 reducing burnout.

→ Such measures are also important for the management of nursing resources and nursing quality [15] as well as for hospital management and administration.

-the concept of affirmative

(2page 29line)

Originating from psychology, affirmative organizational behavior is closely tied  to positive psychological capital, which comprises sub-concepts such as self-efficacy, hope, optimism, and resilience.

→ Originating from psychology, the concept of affirmative organizational behavior is closely tied  to positive psychological capital, which comprises sub-concepts such as self-efficacy, hope, optimism, and resilience.

-in the traditional sense

(2page 33line)

This emphasizes the importance of the positive psychological values of an individual, 33 which may produce more than traditional capital could contribute [16,18].

→ This emphasizes the importance of the positive psychological values of an individual, 33 which may produce more than capital in the traditional sense could contribute [16,18].

(3page 4line)

Based on the above review, we identified we focus on five key areas in this study:

→ Based on the above review, we identified five key areas in this study:

- Holding which variables constant?

(3page 2line)

This study aimed to investigate the levels of workplace bullying, positive psychological capital, and social support and their relationship with burnout among Korean clinical nurses. Based on the above review, we identified five key areas in this study: (1) identifying the general characteristics of Korean clinical nurses, (2) identifying the degree of burnout in relation to the general characteristics of Korean clinical nurses, (3) examining the extent of burnout, workplace bullying, positive psychological capital, and social support among Korean clinical nurses, (4) analyzing the correlations between burnout and bullying, positive psychological capital, and social support, and (5) examining the extent to which burnout is predicted by bullying, positive psychological capital, and social burnout among Korean clinical nurses.

→ This study aimed to investigate the levels of workplace bullying, positive psychological capital, and social support and their relationship with burnout among Korean clinical nurses. Based on the above review, we identified five key areas in this study: (1) identifying the degree of burnout in relation to the demographical characteristics of Korean clinical nurses, (2) examining the level of burnout, workplace bullying,  positive psychological capital, and social support among Korean clinical nurses, (3) analyzing the correlations between burnout and bullying, positive psychological capital, and social support, and (4) examining the level to which burnout is predicted by bullying, positive psychological capital, and social support among Korean clinical nurses.

- Maybe move to the end of this paragraph

(3page 31line)

In total, 180 questionnaires were distributed, of which 175 copies were collected, with a recovery ratio of 97.2%. Excluding 9 questionnaires with insufficient responses, 166 copies were used for the final analysis. Clinical nurses were asked to provide informed consent prior to participating in the study. The sample size was calculated using G*power 3.1.2 [29] and was determined to be 147 to maintain a significance level of .05, an intermediate effect size of .15, and statistical power of .90. Considering the likely attrition rate, 180 questionnaires were distributed. Among them, 175 were returned (response rate = 97.2%) and 166 (excluding 9 incomplete questionnaires) were analyzed.

→ The sample size was calculated using G*power 3.1.2 [29] and was determined to be 147 to maintain a significance level of .05, an intermediate effect size of .15, and statistical power of .90. Considering the likely attrition rate, 180 questionnaires were distributed. In total, 180 questionnaires were distributed, of which 175 copies were collected, with a recovery ratio of 97.2%. Excluding 9 questionnaires with insufficient responses, 166 copies were used for the final analysis. Clinical nurses were asked to provide informed consent prior to participating in the study.

-Please make clear which subscale each example item belongs to.

(3page 43line)

Example items regarding the subscale that measured burnout are: "I get tired after work," "It is annoying to perform nursing work," and "I want to leave the nursing job."

→ Example items regarding the subscale that measured burnout are: "I get tired after work (physical burnout)," "It is annoying to perform nursing work (emotional burnout)," and "I want to leave the nursing job (mental burnout)."

-You only provide one example question.

-It would make more sense to provide one example for each subscale.

-space no capital v

- I disagree that this can be defined as bullying per se.

Thank you so much for your feedback. I added an example. Also, I revised an example.

(4page 11line)

Example questions for the bullying in the workplace subscale included(Verbal and nonverbal bullying): "I have heard disgusting vulgarities or swearing words from other coworkers several times."

→ Example questions for the bullying in the workplace subscale included (Verbal and nonverbal bullying): “I was publicly and often ignored by the other person.”, “The other person suddenly threw something at me or around me.”

- It would make more sense to provide one example for each subscale.

(4page 37line)

Example items 36 that measured social support were as follows: "My direct supervisor listens to my stories about problems related to my work" and "If something difficult occurs while working, I can rely on my direct supervisor."

→ Example items 36 that measured social support were as follows (Support of boss, Support of colleague): "If something difficult occurs while working, I can rely on my direct supervisor." And “I have a colleague who is very close to me.”

(8page 6line)

  1. Discussion

This study was conducted to investigate the associations between bullying, positive psychological capital, and social support and whether these factors predicted burnout in Korean clinical nurses. The main factors for the burnout of clinical nurses were deter-mined by inputting the variables into the model in the order of workplace bullying, posi-tive psychological capital, and social support and analyzing increases in explanatory 11 power. The main purpose of this study was to empirically investigate the associations be-tween bullying, positive psychological capital, and social support and whether they pre-dicted burnout while controlling for general characteristics.

→ 4. Discussion

The main purpose of this study was to empirically investigate the associations be-tween bullying, positive psychological capital, and social support and whether they pre-dicted burnout while controlling for general characteristics.

-Remove causal language: nurses with more positive psychological capital tend to have less burnout.

(8page 11line)

That is, with more workplace 17 bullying, burnout is higher while positive psychological capital leads to lower burnout.

→ That is, with more workplace 17 bullying, burnout is higher while positive psychological capital tends to have less burnout.

-Which are? This is important for an international audience to know.

Thank you so much for your feedback. I revised the sentence.

(8page 19line)

This difference may be due to changes in the perceptions of the participants, reflecting the current trends in society, hospital size, and regional and organizational culture.

There is a discrepancy and that may well be because of perceptions of the participants, reflecting the current trends in society, hospital size, and regional and organizational culture.

-What does this mean? I do not understand.

Thank you so much for your feedback. I revised the sentence.

(9page 18line)

As the degree of burnout found in this study is within the range of 3 to 4 points on the Tedium Scale pro- posed by Pines, Aronson, and Kafry [31], this indicates that careful inspection of nurses’ workplace is required to determine whether the environment and priority of nursing can be changed.

→ As the degree of burnout found in this study is within the range of 3 to 4 points on the Tedium Scale proposed by Pines, Aronson, and Kafry [31], According to previous studies, it is a high figure that requires intervention.

Reviewer 2 Report

Dear Authors, 

Thank you for allowing me to review your manuscript. I believe the article has been improved. 

Best regards.

Author Response

Point 1: Dear Authors, 

Thank you for allowing me to review your manuscript. I believe the article has been improved. 

Best regards.

Response 1: Thank you so much for your feedback. Again, thank you for your feedback.

Reviewer 3 Report

This manuscript examined nurses' burnout predicted by workplace bullying, positive psychological capital, and social support.

It is an interesting issue but requires some revisions

Study purposes need revision: purposes (1) is unnecessary and (2) should be more specified (e.g. general characteristics --> demographical characteristics?). Also purpose (3) is unclear; What is 'the extent of burnout, workplace bullying,  positive psychological capital, and social support'?)

What is uniqueness of this study? Given the previous literature, it is likely to be assumed that the independent variables are negative related(influenced) to burnout(DV). The needs, and strengths of the study could be more clarified and emphasized.

Results

Table 1. The title of the table could be more clarified; the table includes Means and SDs of burnout(?) by respondents' demographic information and workplace bullying, positive psychological capital, and social support? Also, the M and SDs shown by participants' characteristic seem to indicate those of burnout, but that was not indicated (so, it is unclear what those scores mean)

Across tables, the significant levels need to be consistent; in table 3, they were indicated as <.001, but in table 4 they were shown as  .001, <.001* (?)

Discussion

The authors seem to put much efforts on organizing previous studies to support the results of the study. This is very respectable.  However, to emphasize the strength of the study, some interesting results need be interpreted and discussed. For example, in table4, nurses working under 3shift system predicted their burnout levels significantly, but social support did not. These were quite interesting but not discussed. Similarly, age did not predict burnout level, but (considering table 1, nurses who were younger and with lower career experienced more burnout than older group, and higher career). These can be discussed in discussion section.

Page.11, Line 9-19. The reason and interpretation of the results of social support was a unsignificant predictor but workplace bullying was significant to predict burnout. Carefully assumed, psychological capital itself may not have influenced burnout directly, rather it may have indirect effect: that is, the psychological capital may mediate the relation between bullying and burnout. This is just my personal assumption impressed by the results of the manuscript. (Also, workplace bullying is less likely to happen when social support is firmly established. Thus, it may be possible that social support may influence bullying, which may predict burnout). It would be great if the results could be interpreted more fruitfully.

Nurses' burnout is a very important issue, especially it is more so under COVID 19.  Hope some of the comments are helpful for revising manuscript.

Author Response

Point 1: This manuscript examined nurses' burnout predicted by workplace bullying, positive psychological capital, and social support.

It is an interesting issue but requires some revisions

Study purposes need revision: purposes (1) is unnecessary and (2) should be more specified (e.g. general characteristics --> demographical characteristics?). Also purpose (3) is unclear; What is 'the extent of burnout, workplace bullying, positive psychological capital, and social support'?)

Response 1: Thank you so much for your feedback. I revised the sentence. Again, thank you for your feedback.

(3page 2line)

This study aimed to investigate the levels of workplace bullying, positive psychological capital, and social support and their relationship with burnout among Korean clinical nurses. Based on the above review, we identified five key areas in this study: (1) identifying the general characteristics of Korean clinical nurses, (2) identifying the degree of burnout in relation to the general characteristics of Korean clinical nurses, (3) examining the extent of burnout, workplace bullying, positive psychological capital, and social support among Korean clinical nurses, (4) analyzing the correlations between burnout and bullying, positive psychological capital, and social support, and (5) examining the extent to which burnout is predicted by bullying, positive psychological capital, and social burnout among Korean clinical nurses.

→ This study aimed to investigate the levels of workplace bullying, positive psychological capital, and social support and their relationship with burnout among Korean clinical nurses. Based on the above review, we identified five key areas in this study: (1) identifying the degree of burnout in relation to the demographical characteristics of Korean clinical nurses, (2) examining the level of burnout, workplace bullying,  positive psychological capital, and social support among Korean clinical nurses, (3) analyzing the correlations between burnout and bullying, positive psychological capital, and social support, and (4) examining the level to which burnout is predicted by bullying, positive psychological capital, and social support among Korean clinical nurses.

Point 2: What is uniqueness of this study? Given the previous literature, it is likely to be assumed that the independent variables are negative related(influenced) to burnout(DV). The needs, and strengths of the study could be more clarified and emphasized.

Response 2: We were grateful for this opportunity to correct the paper. Thank you so much for your feedback. As you said, as a result of previous studies, burnout and independent variables are negative. This study has shown support for workplace bullying and positive psychological capital predicting burnout. The strength of this study is while social support was a positive external factor that provided help to individuals, it did not significantly predict burnout. In other words, this suggests that internal factors, such as positive psychological capital, are more important than external factors. Thus, as individuals with higher positive psychological capital showed higher levels of environmental factor awareness, each institution should seek to develop educational programs to promote positive competencies. Carefully planned and continued monitoring against workplace bullying will also help create a healthier organizational culture. Again, thank you for your feedback.

Point 3: Results

Table 1. The title of the table could be more clarified; the table includes Means and SDs of burnout(?) by respondents' demographic information and workplace bullying, positive psychological capital, and social support? Also, the M and SDs shown by participants' characteristic seem to indicate those of burnout, but that was not indicated (so, it is unclear what those scores mean)

Response 3: Thank you so much for your feedback. I revised the table. Again, thank you for your feedback.

(5page 17line)

Table 1. Demographic characteristics and main variables

(N=166)

Characteristics

Category

n (%)

M±SD

t/F

p

Scheffe

Sex

Female

155(93.4)

3.36±0.59

2.02

.045

Male

11(6.6)

2.98±0.72

Age

29.8±6.8

7.31

<.001*

≤25

60(36.1)

3.47±0.58

b

26∼30

47(28.3)

3.40±0.55

b

31∼35

29(17.5)

3.40±0.57

b

≥36

30(18.1)

2.89±0.60

a

Clinical

Career(yrs)a

7.02±6.82

7.02

<.001*

≤1

15(9.0)

3.31±0.59

ab

1< ≤5

70(42.2)

3.48±0.62

b

5< ≤10

43(25.9)

3.41±0.49

b

>10

38(22.9)

2.97±0.58

a

Education Level

Junior College Grad

59(35.5)

3.49±0.58

12.18

<.001*

b

University Grad

79(47.6)

3.38±0.54

b

Graduate School or More

28(16.9)

2.86±0.62

a

Religion

Religious

125(75.3)

3.30±0.63

-1.19

.236

Atheistic

41(24.7)

3.43±0.53

Marital status

Unmarried

121(72.9)

3.44±0.57

-3.91

<.001*

Married

45(27.1)

3.04±0.62

Position

Junior Nurse

144(86.7)

3.43±0.56

5.87

<.001*

Staff or Higher

22(13.3)

2.69±0.52

Working

Harassment Experience

Experience of Witnessing Bullying

2 Shift

31(18.7)

3.11±0.53

18.48

2.45

1.70

<.001*

.015

.091

b

3 Shift

120(72.3)

3.47±0.54

c

Regular

15(9.0)

2.64±0.70

a

Yes

62(37.3)

3.48±0.57

No

104(62.7)

3.24±0.61

Yes

82(49.4)

3.41±0.60

Working

Harassment

Severity

Perception

No

84(50.6)

3.25±0.60

18.48

2.87

<.001*

.038

Not severe at all

31(18.7)

3.36±0.51

Harassment Experience

Not Severe

97(58.4)

3.25±0.62

2.45

.015

Severe

34(20.5)

3.44±0.58

Experience of Witnessing Bullying

Very Severe

4(2.4)

3.33±0.61

1.70

.091

Table 1. Demographic characteristics according to the burnout of participants.

(N=166)

Characteristics

Category

n (%)

M±SD

t/F

p

Scheffe

Sex

Female

155(93.4)

3.36±0.59

2.02

.045

Male

11(6.6)

2.98±0.72

Age

29.8±6.8

7.31

<.001

≤25

60(36.1)

3.47±0.58

b

26∼30

47(28.3)

3.40±0.55

b

31∼35

29(17.5)

3.40±0.57

b

≥36

30(18.1)

2.89±0.60

a

Clinical

Career(yrs)a

7.02±6.82

7.02

<.001

≤1

15(9.0)

3.31±0.59

ab

1< ≤5

70(42.2)

3.48±0.62

b

5< ≤10

43(25.9)

3.41±0.49

b

>10

38(22.9)

2.97±0.58

a

Education level

Junior College Grad

59(35.5)

3.49±0.58

12.18

<.001

b

University Grad

79(47.6)

3.38±0.54

b

Graduate School or More

28(16.9)

2.86±0.62

a

Religion

Religious

125(75.3)

3.30±0.63

-1.19

.236

Atheistic

41(24.7)

3.43±0.53

Marital status

Unmarried

121(72.9)

3.44±0.57

-3.91

<.001

Married

45(27.1)

3.04±0.62

Position

Junior Nurse

144(86.7)

3.43±0.56

5.87

<.001

Staff or Higher

22(13.3)

2.69±0.52

Working style

2 Shift

31(18.7)

3.11±0.53

18.48

<.001

b

3 Shift

120(72.3)

3.47±0.54

c

Regular

15(9.0)

2.64±0.70

a

Bullying experience

Yes

62(37.3)

3.48±0.57

2.45

.015

No

104(62.7)

3.24±0.61

Experience of witnessing bullying

Yes

82(49.4)

3.25±0.60

1.70

.091

No

84(50.6)

3.36±0.51

Bullying severity perception

Not severe at all

31(18.7)

3.36±0.51

2.87

.038

Not Severe

97(58.4)

3.25±0.62

Severe

34(20.5)

3.44±0.58

Very Severe

4(2.4)

3.33±0.61

Point 4: Across tables, the significant levels need to be consistent; in table 3, they were indicated as <.001, but in table 4 they were shown as  .001, <.001* (?)

Response 4: Thank you so much for your feedback. I revised the table. Again, thank you for your feedback.

(5page 17line)

Table 1. Demographic characteristics and main variables

(N=166)

Characteristics

Category

n (%)

M±SD

t/F

p

Scheffe

Sex

Female

155(93.4)

3.36±0.59

2.02

.045

Male

11(6.6)

2.98±0.72

Age

29.8±6.8

7.31

<.001*

≤25

60(36.1)

3.47±0.58

b

26∼30

47(28.3)

3.40±0.55

b

31∼35

29(17.5)

3.40±0.57

b

≥36

30(18.1)

2.89±0.60

a

Clinical

Career(yrs)a

7.02±6.82

7.02

<.001*

≤1

15(9.0)

3.31±0.59

ab

1< ≤5

70(42.2)

3.48±0.62

b

5< ≤10

43(25.9)

3.41±0.49

b

>10

38(22.9)

2.97±0.58

a

Education Level

Junior College Grad

59(35.5)

3.49±0.58

12.18

<.001*

b

University Grad

79(47.6)

3.38±0.54

b

Graduate School or More

28(16.9)

2.86±0.62

a

Religion

Religious

125(75.3)

3.30±0.63

-1.19

.236

Atheistic

41(24.7)

3.43±0.53

Marital status

Unmarried

121(72.9)

3.44±0.57

-3.91

<.001*

Married

45(27.1)

3.04±0.62

Position

Junior Nurse

144(86.7)

3.43±0.56

5.87

<.001*

Staff or Higher

22(13.3)

2.69±0.52

Working

Harassment Experience

Experience of Witnessing Bullying

2 Shift

31(18.7)

3.11±0.53

18.48

2.45

1.70

<.001*

.015

.091

b

3 Shift

120(72.3)

3.47±0.54

c

Regular

15(9.0)

2.64±0.70

a

Yes

62(37.3)

3.48±0.57

No

104(62.7)

3.24±0.61

Yes

82(49.4)

3.41±0.60

Working

Harassment

Severity

Perception

No

84(50.6)

3.25±0.60

18.48

2.87

<.001*

.038

Not severe at all

31(18.7)

3.36±0.51

Harassment Experience

Not Severe

97(58.4)

3.25±0.62

2.45

.015

Severe

34(20.5)

3.44±0.58

Experience of Witnessing Bullying

Very Severe

4(2.4)

3.33±0.61

1.70

.091

(8page 1line)

Table 4. Influencing factors on burnout of Korean clinical nurses.

(N=166)

Variables

Model 1

Model 2

Model 3

Model 4

β(p)

β(p)

β(p)

β(p)

General Characteristics

Age

-.02(.905)

.06(.604)

.13(.251)

.10(.387)

Junior College Graduate

.07(.651)

.11(.436)

.09(.466)

.08(.518)

University Undergraduate

-.03(.840)

.02(.879)

.02(.909)

.00(1.000)*

Marital Status

.11(.265)

.09(.358)

.07(.475)

.07(.421)

Position§

.20(.080)

.19(.079)

.12(.259)

.11(.306)

2 Shifts

.16(.168)

.14(.204)

.15(.142)

.16(.130)

3 Shifts

.37(.005)

.38(.002)

.41(.001)

.40(.001)*

Workplace Bullying

.33(<.001)

.26(<.001)

.24(.001)*

Positive Psychological Capital

-.31(<.001)

-.28(<.001)*

Social Support

-.08(.286)

F(p)

7.47(<.001)

10.39(<.001)

12.25(<.001)

11.25(<.001)*

R2

.276

.375

.441

.445

Adj. R2

.239

.339

.405

.406

Adj. R2 change

.239

.100

.066

.001

Note. S.E (Standard Error), β (Standardized Coefficient), R2 (R Squared), Adj.R2 (Adjusted R     Squared); Dummy variables: Dummy variables: Male, Graduate School or More, Married, Staff Nurse or Higher, regular.

Table 1. Demographic characteristics according to the burnout of participants            

(N=166)

Characteristics

Category

n (%)

M±SD

t/F

p

Scheffe

Sex

Female

155(93.4)

3.36±0.59

2.02

.045

Male

11(6.6)

2.98±0.72

Age

29.8±6.8

7.31

<.001

≤25

60(36.1)

3.47±0.58

b

26∼30

47(28.3)

3.40±0.55

b

31∼35

29(17.5)

3.40±0.57

b

≥36

30(18.1)

2.89±0.60

a

Clinical

Career(yrs)a

7.02±6.82

7.02

<.001

≤1

15(9.0)

3.31±0.59

ab

1< ≤5

70(42.2)

3.48±0.62

b

5< ≤10

43(25.9)

3.41±0.49

b

>10

38(22.9)

2.97±0.58

a

Education level

Junior College Grad

59(35.5)

3.49±0.58

12.18

<.001

b

University Grad

79(47.6)

3.38±0.54

b

Graduate School or More

28(16.9)

2.86±0.62

a

Religion

Religious

125(75.3)

3.30±0.63

-1.19

.236

Atheistic

41(24.7)

3.43±0.53

Marital status

Unmarried

121(72.9)

3.44±0.57

-3.91

<.001

Married

45(27.1)

3.04±0.62

Position

Junior Nurse

144(86.7)

3.43±0.56

5.87

<.001

Staff or Higher

22(13.3)

2.69±0.52

Working style

2 Shift

31(18.7)

3.11±0.53

18.48

<.001

b

3 Shift

120(72.3)

3.47±0.54

c

Regular

15(9.0)

2.64±0.70

a

Bullying experience

Yes

62(37.3)

3.48±0.57

2.45

.015

No

104(62.7)

3.24±0.61

Experience of witnessing bullying

Yes

82(49.4)

3.25±0.60

1.70

.091

No

84(50.6)

3.36±0.51

Bullying severity perception

Not severe at all

31(18.7)

3.36±0.51

2.87

.038

Not Severe

97(58.4)

3.25±0.62

Severe

34(20.5)

3.44±0.58

Very Severe

4(2.4)

3.33±0.61

Table 4. Influencing factors on burnout of Korean clinical nurses.          

 (N=166)

Variables

Model 1

Model 2

Model 3

Model 4

β(p)

β(p)

β(p)

β(p)

General characteristics

Gender

.18(.010)

.21(.002)

.18(.004)

.18(.003)

Age

-.02(.905)

.06(.604)

.13(.251)

.10(.387)

Junior college graduate

.07(.651)

.11(.436)

.09(.466)

.08(.518)

University undergraduate

-.03(.840)

.02(.879)

.02(.909)

.00(1.000)

Marital status

.11(.265)

.09(.358)

.07(.475)

.07(.421)

Position§

.20(.080)

.19(.079)

.12(.259)

.11(.306)

2 Shifts

.16(.168)

.14(.204)

.15(.142)

.16(.130)

3 Shifts

.37(.005)

.38(.002)

.41(.001)

.40(.001)

Workplace bullying

.33(<.001)

.26(<.001)

.24(.001)

Positive psychological capital

-.31(<.001)

-.28(<.001)

Social support

-.08(.286)

F(p)

7.47(<.001)

10.39(<.001)

12.25(<.001)

11.25(<.001)

R2

.276

.375

.441

.445

Adj. R2

.239

.339

.405

.406

Adj. R2 change

.239

.100

.066

.001

Note. S.E (Standard Error), β (Standardized Coefficient), R2 (R Squared), Adj.R2 (Adjusted R     Squared); Dummy variables: Dummy variables: Male, Graduate School or More, Married, Staff Nurse or Higher, regular.

Point 5: Discussion

The authors seem to put much efforts on organizing previous studies to support the results of the study. This is very respectable.  However, to emphasize the strength of the study, some interesting results need be interpreted and discussed. For example, in table4, nurses working under 3shift system predicted their burnout levels significantly, but social support did not. These were quite interesting but not discussed. Similarly, age did not predict burnout level, but (considering table 1, nurses who were younger and with lower career experienced more burnout than older group, and higher career). These can be discussed in discussion section.

Response 5: We were grateful for this opportunity to correct the paper. Thank you so much for your feedback. As stated below, I added the sentence. Again, thank you for your feedback.

(9page 23line)

Age did not predict burnout level, but nurses who were younger and with lower careers experienced more burnout than the older group, and higher careers. This is the same result as in study Han, Yang, and Yom [3], and the work difficulties and work responsibilities experienced by nurses with low years of experience are higher than the burnout of experienced nurses. It is thought that as clinical experience increases, work proficiency and emotional stability increase, resulting in lower burnout.

Point 6: Page.11, Line 9-19. The reason and interpretation of the results of social support was a unsignificant predictor but workplace bullying was significant to predict burnout. Carefully assumed, psychological capital itself may not have influenced burnout directly, rather it may have indirect effect: that is, the psychological capital may mediate the relation between bullying and burnout. This is just my personal assumption impressed by the results of the manuscript. (Also, workplace bullying is less likely to happen when social support is firmly established. Thus, it may be possible that social support may influence bullying, which may predict burnout). It would be great if the results could be interpreted more fruitfully.

Response 6: I am grateful for this opportunity to correct the paper. Thank you so much for your feedback. I think you're right. Even though the research on the relationship between workplace bullying and burn-out is still insufficient, there has been progress. The relationship between burnout and social support was found to be negatively correlated in several previous studies. This suggests that although social sup-port is an external factor in the relationship between people, it is more important to establish a relationship that helps people promote their internal positive competence by embracing their surrounding environment in a positive manner and inducing internal motivation, as well as promoting self-efficacy. As social support may vary according to the surrounding environment, it is necessary to find a way to maximize the moderation effect of social support by developing the positive capacity of the individual. Again, thank you for your feedback.

This manuscript is a resubmission of an earlier submission. The following is a list of the peer review reports and author responses from that submission.

Round 1

Reviewer 1 Report

The authors have made some important progress with this manuscript compared to its last version. Unfortunately, I still cannot recommend the publication of the manuscript in its current form. The manuscript still has several important flaws:

  1. There is still a lot of causal language with regard to the analysis of this study. However, the study is correlational, so this causal language is misleading. It also suggests that the literature review might be partially misrepresented (as proving causality when in fact the analyses only show a correlation).
  2. Several sections in the manuscript are difficult to understand, apparently due to a lack of English skills. I strongly recommend that a native speaker edits the manuscript.
  3. There are a good number of typos and formatting issues. Please correct to avoid the impression of sloppy work.
  4. The same results are presented in tables and in text. This is redundant.
  5. Measurements still need more information/clarification to get a better sense of the analyses.
  6. more comments: please see the attachment
